# Confirming validity of The Fear of COVID-19 Scale in Japanese with a nationwide large-scale sample

Haruhiko Midorikawa[1], Miyuki Aiba[2], Adam Lebowitz[3], Takaya Taguchi[4], Yuki Shiratori[5], Takafumi Ogawa[1], Asumi Takahashi[1], Sho Takahashi[4], Kiyotaka Nemoto[6], Tetsuaki Arai[6], Hirokazu Tachikawa[4]*

1 Majors of Clinical Sciences, Graduate School of Comprehensive Human Sciences, University of Tsukuba, Tennoudai, Tsukuba, Ibaraki, Japan, 2 Faculty of Human Sciences, Toyo Gakuen University, Hongo, Bunkyo, Tokyo, Japan, 3 General Studies Department, Jichi Medical University, Yakushiji, Shimotsuke, Tochigi, Japan, 4 Division of Clinical Medicine, Department of Disaster and Community Psychiatry, Faculty of Medicine, University of Tsukuba, Tennoudai, Tsukuba, Ibaraki, Japan, 5 Department of Psychiatry, Tsukuba University Health Center, Tennoudai, Tsukuba, Ibaraki, Japan, 6 Department of Psychiatry, Faculty of Medicine, University of Tsukuba, Tennoudai, Tsukuba, Ibaraki, Japan

* tachikawa@md.tsukuba.ac.jp

**Data Availability Statement:** All relevant data are within the paper and its Supporting Information files.

## Abstract

Assessing fear and anxiety regarding COVID-19 viral infection is essential for investigating mental health during this epidemic. We have developed and validated a Japanese-language version of The Fear of COVID-19 Scale (FCV-19S) based on a large, nationwide residential sample (n = 6,750) recruited through news and social media responding to an online version of the questionnaire. Data was collected from August 4–25, 2020. Results correlated with K6, GAD-7 and IES-R psychological scales, and T-tests and analysis of variance identified associated factors. All indices indicated the two-factor model *emotional fear reactions* and *symptomatic expressions of fear* a better fit for our data than a single-factor model in Confirmatory Factor Analysis ($\chi^2$ = 164.16, p<0.001, CFI 0.991, TLI = 0.985, RMSEA = 0.043). Socio-demographic factors identified as disaster vulnerabilities such as female sex, sexual minority, elderly, unemployment, and present psychiatric history associated with higher scores. However, respondent or family member experience of infection risk, or work/school interference from confinement, had greatest impact. Results suggest necessity of mental health support during this pandemic similar to other disasters.

## Introduction

Corona virus disease 2019 (COVID-19) has become an important and urgent threat to global health. Since the cluster of cases of pneumonia of unknown etiology were reported in Wuhan, China in December 2019 [1], COVID-19 transmission continued spreading, and on 30 January 2020 WHO declared the outbreak a Public Health Emergency of International Concern [2]. Subsequently, despite various public health responses aimed at slowing the spread of COVID-19, many countries have faced a critical health crisis [3]. As of early September 2020,

**Funding:** Program to Apply the Wisdom of the University to tackle COVID -19 Related Emergency Problems. (https://www.osi.tsukuba.ac.jp/fight_covid19/) The funders had no role in study design, data collection and analysis, decision to publish, or preparation of the manuscript.

**Competing interests:** The authors have declared that no competing interests exist.

25 million cases have been identified and more than 800,000 deaths have occurred [4]. While various treatments are being practiced and researched [5], the impact of COVID-19 is expected to continue [6].

In Japan, by early September 2020 more than 7,000 infections had been identified, with about 1,400 COVID-related deaths. The first wave of infections began March-April 2020, and the Japanese government declared a state of emergency through May 25 [7]. During that time, a mild lockdown was implemented relying on voluntary cooperation [8]. This lockdown was not accompanied by any legal penalties. Although prefectural governors could only request people refrain from going out unnecessarily, people's activities were curbed to some extent and the number infections did not explode. Later, due to the worsening economic situation and increasing number of suicides [9], the national government started to subsidize travel and dining out from summer to encourage resumption of socioeconomic activities. However, the second wave of infection arrived almost simultaneously, and infections have been gradually increasing.

COVID-19 causes psychological as well as physical problems [10, 11]. The main psychological impact of the spread of infection is elevated rates of stress, anxiety, depression and frustration [12]. In addition, rising levels of loneliness, depression, harmful alcohol and drug use, and self-harm or suicidal behavior are also expected [13]. Psychological problems appeared not only in those infected but also in health care workers [14, 15]. These problems are due to fears of disease and to mitigation policies in many countries such as lockdowns, quarantines, and physical distancing [16].

Assessing anxiety about COVID-19 is important in investigating people's mental health during the epidemic. Underlying behaviors negatively affecting mental health related to the infection, such as prejudice, discrimination, and stigmatization, come from anxiety and fear related to the infection [17, 18]. On the other hand, such feelings are also a normal response to a life-threatening situation. It has been suggested anxiety and fear play an important role in motivating adherence to preventive behaviors (e.g., social distancing, improved hand hygiene) [19, 20].

The Fear of COVID-19 Scale (FCV-19S) was developed to measure anxiety and fear of COVID-19 [21]. FCV-19S is a simple seven-item self-administered questionnaire and has been translated and validated in a number of countries [22–29]. COVID-19 is not just an infectious disease, but has serious social impacts. Since COVID-19 response differs between societies, it is important to examine psychometric characteristics of the scale for each country, rather than simply compare responses between translated versions. Actually, in terms of factor structure, some studies support a one-factor while others support a two-factor model. In Japan, two studies have validated the instrument [30, 31]. However, despite the fact viral impact varies by age [32], one study [30] tested only a small number of students. The other [31] included adults participants but only examined validity of a one-factor model. In addition, large-scale studies have only been conducted in a few countries. Therefore, to overcome previous shortcomings when examining psychometric properties and clarify mental health impacts of COVID-19 in Japan, we conducted a large-scale Japanese-language validation (FCV-19S-J).

## Method

### Survey method

The survey was conducted among Japanese residing in Japan. Participants were widely recruited through several news media and social media (Twitter and Facebook). The survey was administered to those who agreed to participate. Data was collected from August 4–25, 2020 with the online platform SurveyMonkey [33]. SurveyMonkey is an online survey service

that facilitates sharing surveys via email, smartphone applications, and social media platforms such as Facebook and Twitter. Exclusion criteria for participants were: second or later responses with a duplicate IP address, and no response. The number of participants was confirmed to be sufficient by referring to the previous study on FCV-19S [21] and literature that indicated the minimum sample size required for CFA [34].

## Questionnaire

FCV-19S is a seven-item self-administered scale developed by Ahorsu et al. [21]. Answers included "strongly disagree," "disagree," "neither agree nor disagree," "agree," and "strongly agree". The minimum score possible for each question is 1 (strongly disagree), and the maximum is 5 (strongly agree). A total score is calculated by adding up each item score (ranging from 7 to 35). The higher the score, the greater the fear of COVID-19. We created the Japanese version of the FCV-19S by referring to the guidelines of the International Society for Pharmacoeconomics and Outcomes Research Task Force [35]. Two psychiatrists independently translated the original version of the FCV-19S from English to Japanese, with the permission of the original author. Both translations were integrated into a single version back-translated into English by a native speaker literate in Japanese and reviewed by the research team.

The questionnaire also included socio-demographic items (gender, age group, occupation, residence, and history of psychiatric treatment), and psychological scales: FCV-19S-J, Kessler Screening Scale for Psychological Distress (K6), Generalized Anxiety Disorder -7 (GAD-7), and Impact of Event Scale-Revised (IES-R). The K6 is a short, six-item questionnaire developed to screen for mood and anxiety disorders. The total score of the K6 ranges from 0–24. We used the reliable and validated Japanese version of the K6 in the survey [36]. The GAD-7 is a self-administered questionnaire developed to assess the severity of Generalized Anxiety Disorder (GAD) by extracting questions related to anxiety disorders from the PHQ [37]. It consists of seven items, and symptom intensity during the past two weeks is rated on a 4-point scale (0 to 3) with a total score ranging from 0–21. We used the reliable and validated Japanese version of the GAD-7 in the survey [38]. The IES-R (Impact of Event Scale-Revised), a revision of the former IES [39], is a self-administered questionnaire developed to investigate traumatic distress [40]. The IES-R consists of 22 items, including 7 intrusion items, 8 avoidance items, and 7 hyperarousal items, and evaluates the intensity of symptoms in the past week on a 5-point scale (0 to 4). The total score ranges from 0–88. We used the reliable and validated Japanese version of the IES-R for the survey [41].

Finally, question items tapped experience during the COVID-19 epidemic. A stress level question "How stressful have you felt over the past month in relation to the COVID-19?" had Likert scale responses: "not at all (1)," "not much (2)," "neither (3)," "a little (4)," "very much (5)" and "other (6)". Other times were: "I was at risk of infection," "My family members were at risk of infection," "I was home alone because I was at risk of infection," "Self-confinement interfered with work or school," and "I or my family members have been discriminated against or bullied." Multiple selections were possible, and the first three experiences were combined into one item "I or my family was at risk of infection" in the analysis. Participants were also asked a multiple-choice question about effective coping methods during the self-confinement period.

## Statistical analysis

We first examined the distribution of respondents' socio-demographic characteristics and experience during COVID-19 epidemic. Next, we conducted confirmatory factor analyses (CFA) of single-factor and two-factor model with maximum likelihood estimation to verify

structural validity of Japanese translated FCV-19S-J items and compared goodness of fit:: comparative fit index (CFI ≥ .95), Tucker–Lewis index (TLI ≥ .95), root mean square error of approximation (RMSEA ≤ .06), and Akaike's information criterion (AIC) [42, 43]. Reliability was evaluated using Cronbach's alpha. To examine construct validity, we also calculated Pearson correlation coefficients between the FCV-19S-J and self-assessment mental health scales K6, GAD-7, IES-R. Strength of correlation coefficient values were: $r < 0.30$ "weak," $r = 0.30$–$0.59$ "moderate", and $r \geq 0.60$ "strong" [44]. T-tests and analysis of variance (ANOVA) examined association between FCV-19S-J scores, and socio-demographic and epidemic experience data during COVID-19. P-values were adjusted by Bonferroni correction for multiple comparisons ($p < 0.05/27 = 0.00185$). The percentage of missing values for each item in the socio-demographic characteristics was shown in the characteristics of respondents. Since the number of missing values was very small, we did not consider the presence of missing values in the socio-demographic factor in the factor analysis. In the correlation analysis, T-test, and ANOVA, missing values were excluded for each statistical testing. CFA was performed with the statistical package Amos version 26 for Windows and the other statistical analyses were performed using IBM SPSS version 22 for Windows.

### Ethical consideration

Before starting the survey, we explained the purpose of the survey and that participation was voluntary and that survey was anonymous. Participants gave their consent by ticking a box to confirm that they understood the information provided to them and voluntarily agreed to participate in the survey. Only those who agreed to cooperate in the survey would be able to proceed to the questionnaire. This study was approved by the medical ethics committee of the University of Tsukuba (Registration No.1546-1).

## Results

### Respondents' characteristics and experience during COVID-19 epidemic

A total of 7912 responses were obtained, 7389 after duplicate responses were removed based on IP addresses. The attrition rate was 19.2% (1,420/7,389). Of these, n = 6750 were included in the analysis (no response to FCV-19S-J = 639). 4283 (63.5%) of the respondents were female. Age of respondents ranged from teenagers to over 60 years old, but only 4.3% of the respondents were 60y<. All socio-demographic variables are shown in Table 1.

### Validation of FCV-19S-J

Table 2 reports the factor loadings of single-factor and two-factor models. In the two-factor model, Factor 1 *emotional fear reactions* was composed of psychological dimensions such as anxiety and fear, and Factor 2 *symptomatic expressions of fear* with physiological dimensions such as sweating, palpitations, and insomnia.

The results of the CFA model fit are reported in Table 3. All indices indicated better fit for the two-factor model than a single-factor model. Reliability analysis assessing both models indicated a Cronbach's alpha of 0.83 for the single-factor model, and 0.77 for Factor 1 and 0.83 for Factor 2.

The correlations between FCV-19S-J and other mental health self-assessment measures are shown in Table 4. Total FCV-19S-J*Factor 1 and Total FCV-19S-J*Factor 2 were highly correlated ($r > 0.60$). Total FCV-19S*K6, *GAD-7, and *IES-R were moderately correlated ($0.30 < r < 0.60$), as were correlations between individual factors and the other scales. Because the association levels between FCV-19S-J items and other scales are not disparately dissimilar, this

**Table 1. Characteristics of respondents.**

| Variable | N (%) | Variable | N (%) |
|---|---|---|---|
| Gender | | Psychiatric history | |
| Male | 2,352 (34.8%) | Present | 1,018 (15.1%) |
| Female | 4,283 (63.5%) | Past | 1,677 (24.8%) |
| Other | 115 (1.7%) | Never | 3,952 (58.5%) |
| Age group | | Unknown/Other | 103 (1.5%) |
| -19 | 147 (2.2%) | The extent of the stress associated with COVID-19 | |
| 20–29 | 1,396 (20.7%) | Not at all | 155 (2.3%) |
| 30–39 | 1,879(27.8%) | Not much | 856 (12.7%) |
| 40–49 | 1,869 (27.7%) | Neither | 253 (3.7%) |
| 50–59 | 1,167 (17.3%) | a little | 2,841 (42.1%) |
| 60- | 292 (4.3%) | Very much | 2,577 (38.2%) |
| Occupation | | Unknown / Other | 68 (1.0%) |
| Non health care worker | 4435 (65.7%) | The experiences during COVID-19 epidemic | |
| Health care worker | 832 (12.3%) | I or my family members were at risk of infection. | 3,896 (57.7%) |
| Unemployed | 855 (12.7%) | Self-confinement interfered with work or school. | 3,440 (51.0%) |
| Student | 609 (9.0%) | I or my family members have been discriminated or bullied. | 148 (2.2%) |
| Unknown | 19 (0.3%) | I had effective coping ways during the self-confinement period | 5,734 (84.9%) |
| Living place | | I or my family members were at risk of infection. | 3,896 (57.7%) |
| ≥5 million or more | 4,641 (68.8%) | Self-confinement interfered with work or school. | 3,440 (51.0%) |
| <5 million | 2,056 (30.5%) | I or my family members have been discriminated or bullied. | 148 (2.2%) |
| Unknown | 53 (0.8%) | I had effective coping ways during the self-confinement period | 5,734 (84.9%) |

indicates construct validity; in particular, concurrent validity for the instrument and convergent validity for the factors being measured.

The mean score of FCV-19S was 16.67±4.851 and the mode was 15, which accounted for 8.3% of the total score. The mean score of Factor1 was 12.18±3.490 and the mode was 14, which accounted for 11.6% of the total score. The mean score of Factor 2 was 4.48±1.983 and 50.0% of the respondents answered no to all questions.

## Factors associated with FCV-19S-J

The association between the FCV-19S-J and socio-demographic factors and experience during COVID-19 epidemic is shown in Tables 5 and 6. FCV-19S-J total score, Factor 1 score and Factor 2 scores were significantly different by gender, age group, occupation, and psychiatric

**Table 2. Factor loadings of single-factor and two-factor model.**

| | | 1-factor | 2-factor |
|---|---|---|---|
| 7 | My heart races or palpitates when I think about getting coronavirus-19. | .81 | .86 |
| 6 | I cannot sleep because I'm worrying about getting coronavirus-19. | .76 | .81 |
| 3 | My hands become clammy when I think about coronavirus-19. | .71 | .69 |
| 5 | When watching news and stories about coronavirus-19 on social media, I become nervous or anxious. | .58 | .80 |
| 2 | It makes me uncomfortable to think about coronavirus-19. | .46 | .66 |
| 1 | I am most afraid of coronavirus-19. | .47 | .64 |
| 4 | I am afraid of losing my life because of coronavirus-19. | .46 | .51 |
| | **correlation between factors** | - | .67 |

**Table 3. Model fit indices of CFA.**

|          | χ2     | (df) | p      | CFI   | TLI   | RMSEA | AIC    |
|----------|--------|------|--------|-------|-------|-------|--------|
| 1-factor | 386.25 | 8    | <0.001 | 0.979 | 0.944 | 0.084 | 426.25 |
| 2-factor | 164.15 | 12   | <0.001 | 0.991 | 0.985 | 0.043 | 196.15 |

history. However, all of these effect sizes were small, and may be significant due to large sample size. Females were higher in total and Factor 1 scores, non-identified in Factor 2. Respondents in their 50s and older tended to have higher scores than other age groups. The unemployed scored consistently higher even compared to health care workers. With regard to residency, respondents in a prefecture with a population of 5 million< had higher scores only in total score. Psychiatric history was also significant in total and factored scores, as was infection risk to self or family, and experiencing work/school interruption. Whether respondents or family members had been discriminated or bullied against was not associated with FCV-19S-J scores, nor was effective coping during self-confinement.

## Discussion

We examined validity of the Japanese version of FCV-19S and clarified the factors related to the fear of COVID-19. To our knowledge, this is one of the largest nationwide surveys in Japan. The results of the CFA revealed that two-factor model had a better model fit than one-factor model. In a Japanese sample, Masuyama et al. concluded the translated FCV-19S was composed of two factors [30] and the goodness of fit of the one-factor model was not high in Wakashima's study [31]. Therefore, a two-factor model composed of *emotional fear reactions* and *symptomatic expressions of fear* appears plausible. Similar results have been confirmed in the Spanish version using a sample in Peru and Lima [45], the Russian version of a sample in Russia and Belarus [46], and the Hebrew version among an Israeli population [27]. The names of these two factors were taken from the study by Tzur Bitan et al. [27]. As a result of validation, the two-factor model showed sufficient values in all of the factor loadings, the goodness of fit indices, and Cronbach's alpha. These results, plus correlation levels with other self-assessment scales, confirm the FCV-19S-J has sufficient validity.

The results of this study showed 80% of respondents felt major stress during the COVID-19 epidemic. The average score for FCV-19S was 16.67. The US and New Zealand, Spain, Russia, Israel and Pakistan were distributed between 15–19 points [22, 25–27, 46, 47], and Asian countries such as Iran, Bangladesh and Turkey had scores above 20 [21, 24, 48]. Compared to other countries, the FCV-19S-J score in Japan was comparatively lower, perhaps due to lower disease

**Table 4. Correlation between FCV-19S and each scale.**

| Scale (N, Mean, SD) | Total | Factor 1 | Factor 2 | K6 | GAD-7 | IES-R | Stress |
|---|---|---|---|---|---|---|---|
| FCV-19 Total (n = 6,750, 16.67, 4.851) | - | 0.939** | 0.795** | 0.429** | 0.491** | 0.544** | 0.511** |
| FCV-19S Factor 1 (n = 6,750, 4.48, 1.983) | | - | 0.536** | 0.368** | 0.415** | 0.467** | 0.533** |
| FCV-19S Factor 2 (n = 6,750, 12.18, 3.490) | | | - | 0.402** | 0.472** | 0.511** | 0.313** |
| K6 (n = 6,633, 7.46, 5.659) | | | | - | 0.789** | 0.670** | 0.315** |
| GAD-7 (n = 6,460, 4.85, 4.734) | | | | | - | 0.734** | 0.331** |
| IES-R (n = 5,969, 15.22, 15.524) | | | | | | - | 0.365** |
| The extent of the stress associated with COVID-19 (n = 6,682, 4.02, 1.07) | | | | | | | - |

**p<0.001.

**Table 5. Association between FCV-19S score and socio-demographic factors.**

| Variable | Total | | | Factor 1 | | | Factor 2 | | |
|---|---|---|---|---|---|---|---|---|---|
| | Mean (SD) | P-value | Effect size | Mean (SD) | P-value | Effect size | Mean (SD) | P-value | Effect size |
| Gender | | | | | | | | | |
| Male (n = 2,352) | 15.31 (4.879) | <0.001 | $\eta^2 p = 0.042$ | 11.07 (3.596) | <0.001 | $\eta^2 p = 0.055$ | 4.24 (1.880) | <0.001 | $\eta^2 p = 0.009$ |
| Female (n = 4,283) | 17.40 (4.645) | | | 12.79 (3.262) | | | 4.60 (2.011) | | |
| Other (n = 115) | 17.25 (5.779) | | | 12.28 (3.895) | | | 4.97 (2.433) | | |
| Age group | | | | | | | | | |
| -19 (n = 147) | 16.63 (5.382) | <0.001 | $\eta^2 p = 0.006$ | 12.26 (3.959) | <0.001 | $\eta^2 p = 0.004$ | 4.37 (2.048) | <0.001 | $\eta^2 p = 0.008$ |
| 20–29 (n = 1,396) | 16.26 (5.002) | | | 11.91 (3.694) | | | 4.35 (1.944) | | |
| 30–39 (n = 1,879) | 16.41 (4.787) | | | 12.09 (3.491) | | | 4.32 (1.93) | | |
| 40–49 (n = 1,869) | 16.71 (4.810) | | | 12.18 (3.418) | | | 4.53 (1.999) | | |
| 50–59 (n = 1,167) | 17.35 (4.766) | | | 12.57 (3.307) | | | 4.78 (2.051) | | |
| 60- (n = 292) | 17.32 (4.517) | | | 12.51 (3.264) | | | 4.80 (1.912) | | |
| Occupation | | | | | | | | | |
| Non health care worker (n = 4,435) | 16.59 (4.812) | <0.001 | $\eta^2 p = 0.006$ | 12.13 (3.467) | <0.001 | $\eta^2 p = 0.005$ | 4.46 (1.954) | <0.001 | $\eta^2 p = 0.005$ |
| Health care worker (n = 832) | 16.63 (4.642) | | | 12.24 (3.400) | | | 4.39 (1.898) | | |
| Unemployed (n = 855) | 17.53 (5.058) | | | 12.71 (3.515) | | | 4.82 (2.171) | | |
| Student (n = 609) | 16.02 (4.979) | | | 11.73 (3.673) | | | 4.28 (1.958) | | |
| Living place | | | | | | | | | |
| ≥5 million (n = 4,641) | 16.98 (4.939) | <0.001 | r = 0.041 | 12.39 (3.486) | 0.002 | r = 0.037 | 4.59 (2.080) | 0.006 | r = 0.045 |
| <5 million (n = 2,056) | 16.55 (4.799) | | | 12.11 (3.482) | | | 4.44 (1.933) | | |
| Psychiatric history | | | | | | | | | |
| Present (n = 1,018) | 17.50 (5.268) | <0.001 | $\eta^2 p = 0.007$ | 12.58 (3.553) | <0.001 | $\eta^2 p = 0.003$ | 4.91 (2.327) | <0.001 | $\eta^2 p = 0.012$ |
| Past (n = 1,677) | 16.80 (4.897) | | | 12.22 (3.478) | | | 4.58 (2.000) | | |
| Never (n = 3,952) | 16.38 (4.677) | | | 12.05 (3.465) | | | 4.32 (1.847) | | |

mortality. Distribution of responses was similar, although presentation differs among studies. For example, Soraci et al. showed a higher number of Italian respondents who selected "strongly disagree" for items under Factor 2 (Item 3, Item 6, Item 7) [23]. In the Bangladeshi,

**Table 6. Association between FCV-19S score and experiences during COVID-19 epidemic.**

| Variable | Total | | | Factor 1 | | | Factor 2 | | |
|---|---|---|---|---|---|---|---|---|---|
| | Mean (SD) | P-value | Effect size | Mean (SD) | P-value | Effect size | Mean (SD) | P-value | Effect size |
| I or my family members were at risk of infection. | | | | | | | | | |
| Yes (n = 3,896) | 17.43 (4.802) | <0.001 | r = 0.184 | 12.75 (3.348) | <0.001 | r = 0.199 | 4.68 (2.096) | <0.001 | r = 0.122 |
| No (n = 2,854) | 15.62 (4.722) | | | 11.41 (3.533) | | | 4.21 (1.782) | | |
| Self-confinement interfered with work or school. | | | | | | | | | |
| Yes (n = 3,440) | 17.48 (4.819) | <0.001 | r = 0.171 | 12.75 (3.336) | <0.001 | r = 0.165 | 4.73 (2.123) | <0.001 | r = 0.131 |
| No (n = 3,310) | 15.82 (4.739) | | | 11.60 (3.550) | | | 4.22 (1.789) | | |
| I or my family members have been discriminated or bullied. | | | | | | | | | |
| Yes (n = 148) | 17.21 (5.415) | 0.168 | r = 0.017 | 12.54 (3.789) | 0.207 | r = 0.015 | 4.67 (2.305) | 0.250 | r = 0.014 |
| No (n = 6,602) | 16.65 (4.837) | | | 12.17 (3.483) | | | 4.48 (1.975) | | |
| I had effective coping ways during the self-confinement period. | | | | | | | | | |
| Yes (n = 5,734) | 16.57 (4.823) | 0.246 | r = 0.074 | 12.12 (3.479) | 0.538 | r = 0.039 | 4.45 (1.963) | 0.077 | r = 0.112 |
| No (n = 235) | 17.01 (5.635) | | | 12.28 (3.931) | | | 4.73 (2.328) | | |

Arabic, Italian and Spanish versions, average scores for Item 3, Item 6 and Item 7 tended to be lower [23–25, 49]. These tendencies were consistent with our data. However, the original version of Ahorsu et al.'s data did not show this trend, suggesting the degree to which fear of COVID-19 causes physical responses varies across countries and regions.

The FCV-19S-J scored higher in females and older adults, similar to Bangladesh, Greece and India [24, 50–52], and a People's Republic of China study using a different instrument [53] In addition, a new finding from our study is respondents who selected a gender other than male or female had higher scores, especially in Factor 2 (physical symptoms). Transgender, non-conforming people and gender dysphoria often have mental health problems due to discrimination, prejudice and social inequalities [54, 55]. Social distance policies associated with COVID-19 may worsen these problems by severing their relationships with supportive and affirmative people and organizations [56]. More attention should be paid to this point.

With regard to age, some reports show higher FCV-19S scores in older adults [50], while others do not [24]. Older adults are known to be at higher risk after COVID-19 infection because they have lower immunity and often have chronic diseases [57–59]. For this reason, it is understandable that their fear of COVID-19 is higher than that of other generations. The high scores of the unemployed are consistent with the study in Pakistan [22].

With the report of low socioeconomic status being associated with fear of COVID-19 [27] and the report of suicide due to the economic slump associated with lockdown [60], the mental health of the socially vulnerable people is an important issue during the COVID-19 epidemic. It is also important to note that FCV-19S-J scores among health care workers did not differ significantly from other occupations. Doshi D et al. reported higher FCV-19S scores among health care workers [52]. As the extent of the risk of infection in the site where each health care worker works is not clear, it would be desirable for future research to clarify this point.

Another new finding is the FCV-19S-J score was higher in psychiatric patients. Chang et al. reported that the FCV-19S is useful for measuring fear of COVID-19 even in patients with psychiatric disorders [51]: however, this study did not compare results with those of patients without psychiatric disorders. Some reports suggest that people with mental illness are more likely to have mental health problems during the COVID-19 epidemic [61, 62]. It should be noted that fear of COVID-19 may be heightened in individuals with pre-existing mental health problems.

Compared to factors mentioned so far, presence of experience, such as whether or not respondents or their family were at risk of infection and whether or not self-confinement interfered with work or school showed larger effects. This is reasonable as these are considered to be more direct factors for fear of COVID-19. As for the result that experiences of discrimination and bullying were not related to FCV-19S-J scores, this might be due to only a small number in the current study reporting such experiences.

Socio-demographic factors found to be associated with fear of COVID-19 in this study, such as females, sexual minorities, the elderly, and the unemployed, were consistent with the characteristics of vulnerable populations during disasters. Females, children, adolescents, the poor, the elderly, and those with pre-existing health problems have been identified as vulnerable populations that often experience psychological morbidity as a result of disasters [63, 64]. This association suggests that COVID-19 is a disaster and that utilizing the findings of disaster psychiatry can be useful in the COVID-19 epidemic. In addition, among the factors identified in this study, attention needs to be paid to older age and unemployment in particular, from the perspective of the digital divide [65]. This is because, during in the midst of the COVID-19 epidemic use of online services is growing rapidly [66]; however, some from these groups may not be able to use such services. In addressing mental health issues associated with COVID-19,

greater attention may need to be paid to those who are unable to benefit from digital health-care technologies.

The limitations of the study are as follows. First, sampling bias needs to be considered because anyone was free to participate in this survey. Socio-demographic factors such as gender, age and living place are not representative of the general population of Japan. Also, the possibility that a population with a strong interest in COVID-19 was selected cannot be ruled out. The reason for the large number of respondents with present psychiatric history may be partially due to psychiatrists' use of social media and snowball sampling in recruiting. Second, the situation of infection changed during the study period, and trends of infection varied by region. For this reason, individual responses cannot be assured to be responses to the same situation. Third, because of the cross-sectional design of this study, it was difficult to assess causality.

Despite these limitations, this study is significant as it confirms the validity of FCV-19 using large-scale data and investigates fear of COVID-19 and related factors at the nationwide level. Besides mental health care of those actually affected by COVID-19, support may be considered necessary for those with vulnerable factors our study identified. Further research is needed to determine which populations are more likely to have a heightened fear of COVID-19 and which are more likely to cause physical and mental problems. To this end, it is desirable to conduct longitudinal study by using representatives of the general population.

## Supporting information

**S1 File. Japanese-language version of FCV-19S (FCV-19S-J).**
(PDF)

**S1 Dataset. Anonymized data set.**
(XLSX)

## Author Contributions

**Conceptualization:** Haruhiko Midorikawa, Takaya Taguchi, Yuki Shiratori, Takafumi Ogawa, Asumi Takahashi, Sho Takahashi, Hirokazu Tachikawa.

**Data curation:** Haruhiko Midorikawa, Miyuki Aiba, Takafumi Ogawa, Asumi Takahashi, Hirokazu Tachikawa.

**Formal analysis:** Haruhiko Midorikawa, Miyuki Aiba.

**Funding acquisition:** Hirokazu Tachikawa.

**Investigation:** Haruhiko Midorikawa, Takafumi Ogawa, Hirokazu Tachikawa.

**Methodology:** Haruhiko Midorikawa, Miyuki Aiba, Hirokazu Tachikawa.

**Project administration:** Haruhiko Midorikawa, Hirokazu Tachikawa.

**Resources:** Hirokazu Tachikawa.

**Software:** Hirokazu Tachikawa.

**Supervision:** Adam Lebowitz, Takaya Taguchi, Yuki Shiratori, Takafumi Ogawa, Sho Takahashi, Kiyotaka Nemoto, Tetsuaki Arai, Hirokazu Tachikawa.

**Validation:** Hirokazu Tachikawa.

**Visualization:** Haruhiko Midorikawa, Miyuki Aiba.

**Writing – original draft:** Haruhiko Midorikawa, Miyuki Aiba.

**Writing – review & editing:** Adam Lebowitz, Asumi Takahashi, Kiyotaka Nemoto, Tetsuaki Arai, Hirokazu Tachikawa.

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
