## [Decision Letter · Decision Letter 0]

7 Jan 2021

PONE-D-20-37753

Confirming Validity of The Fear of COVID-19 Scale in Japanese with a Nationwide Large-Scale Sample.

PLOS ONE

Dear Dr. Tachikawa,

Thank you for submitting your manuscript to PLOS ONE. After careful consideration, we feel that it has merit but does not fully meet PLOS ONE’s publication criteria as it currently stands. Therefore, we invite you to submit a revised version of the manuscript that addresses the points raised during the review process.

Thank you for the valuable contribution. One expert in the psychology field has carefully reviewed your submission and he has found merits in your work. However, several minor but necessary amendments should be well taken care of in a resubmission. I look forward to receiving your revision. 

We look forward to receiving your revised manuscript.

Kind regards,

Chung-Ying Lin

Academic Editor

PLOS ONE

Journal Requirements:

2) Please provide additional details regarding participant consent. In the ethics statement in the Methods and online submission information, please ensure that you have specified what type you obtained (for instance, written or verbal, and if verbal, how it was documented and witnessed). If your study included minors, state whether you obtained consent from parents or guardians. If the need for consent was waived by the ethics committee, please include this information.

3) PLOS requires an ORCID iD for the corresponding author in Editorial Manager on papers submitted after December 6th, 2016. Please ensure that you have an ORCID iD and that it is validated in Editorial Manager. To do this, go to ‘Update my Information’ (in the upper left-hand corner of the main menu), and click on the Fetch/Validate link next to the ORCID field. This will take you to the ORCID site and allow you to create a new iD or authenticate a pre-existing iD in Editorial Manager. Please see the following video for instructions on linking an ORCID iD to your Editorial Manager account: https://www.youtube.com/watch?v=_xcclfuvtxQ

4) Please include captions for your Supporting Information files at the end of your manuscript, and update any in-text citations to match accordingly. Please see our Supporting Information guidelines for more information: http://journals.plos.org/plosone/s/supporting-information.

Reviewers' comments:

Reviewer's Responses to Questions

**Comments to the Author**

1. Is the manuscript technically sound, and do the data support the conclusions?

Reviewer #1: Yes

2. Has the statistical analysis been performed appropriately and rigorously? 

Reviewer #1: Yes

3. Have the authors made all data underlying the findings in their manuscript fully available?

Reviewer #1: Yes

4. Is the manuscript presented in an intelligible fashion and written in standard English?

Reviewer #1: Yes

5. Review Comments to the Author

Reviewer #1: I have read the paper with a great interest. The paper has written up well but needs revision.

I have reported my review as following:

Abstract: Sampling procedure is not clear in abstract. Please add study time. Like Bitan et al I would suggest to rename the factors as : Emotional fear reactions and Symptomatic expressions of fear

Introduction: I think that it will be great if the authors can provide some information regarding how Japan react to the COVID-19 (e.g., government' policies, infected cases, deaths). It would be better to justify why a scale should be translated into other languages. Please update your references on recent publication of Fear validation:

For example:

Piqueras, J. A., Gomez-Gomez, M., Marzo, J. C., Gomez-Mir, P., Falco, R., Valenzuela, B., & COVID, P. S. (2020). Validation of the Spanish version of Fear of COVID-19 Scale: Its association with acute stress and coping.

Elemo, Aman Sado, Seydi Ahmet Satici, and Mark D. Griffiths. "The Fear of COVID-19 Scale: Psychometric Properties of the Ethiopian Amharic Version." International Journal of Mental Health and Addiction (2020): 1-12.

Method: sampling procedure, study time are still vague here. Did you used any exclusion criteria? Please provide a justification/ rationale for the sample size. Please also provide the attrition rate. How did you handle with missing data? The method of estimation for the CFA is not clear. Which software did you use for conducting CFA?

Results: please adjust your P-values for multiple comparisons

6. PLOS authors have the option to publish the peer review history of their article (what does this mean?). If published, this will include your full peer review and any attached files.

Reviewer #1: **Yes: **Amir H Pakpour

---

## [Author Response · Author response to Decision Letter 0]

25 Jan 2021

January 26, 2021

Dear Editors:

Please accept for re-submission our manuscript PONE-D-20-37753 titled “Confirming Validity of The Fear of COVID-19 Scale in Japanese with a Nationwide Large-Scale Sample.” 

We thank the reviewer for the valuable suggestions which greatly improved the quality of the manuscript. Responses to all comments are below. Changes to the manuscript are highlighted.

Sincerely, 

Hirokazu Tachikawa, MD, PhD 

Department of Psychiatry, 

Faculty of Medicine, University of Tsukuba, 

1-1-1 Tennoudai, Tsukuba, Ibaraki, 305-8575, Japan. 

tachikawa@md.tsukuba.ac.jp

Answer to Review #1

Abstract: Sampling procedure is not clear in abstract. Please add study time. Like Bitan et al I would suggest to rename the factors as: Emotional fear reactions and Symptomatic expressions of fear

Thank you very much for this suggestion. We have added the details of sampling in the abstract (p.2, ll. 23-25) and renamed the factors as emotional fear reactions and symptomatic expressions of fear (p.2, l. 27). We also added that the names of these factors were taken from the study by Bitan et al (p.15, ll. 223-224).

Introduction: I think that it will be great if the authors can provide some information regarding how Japan react to the COVID-19 (e.g., government' policies, infected cases, deaths). It would be better to justify why a scale should be translated into other languages. Please update your references on recent publication of Fear validation:

Authors greatly appreciate this suggestion to improve the Introduction. We have added a brief note on Japan's response to COVID-19, explained why FCV-19S should be translated into other languages, and updated the references on FCV-19S (p.3 ll. 50-59, p.4 ll.75-78).

Method: sampling procedure, study time are still vague here. Did you used any exclusion criteria? Please provide a justification/ rationale for the sample size. Please also provide the attrition rate. How did you handle with missing data? The method of estimation for the CFA is not clear. Which software did you use for conducting CFA?

Thank you for these comments. We described the sampling procedure in more detail mentioning the exclusion criteria and the justification for the sample size. We also included information on the handling of dropout rates and missing values. The CFA method was maximum likelihood estimation. Information regarding software used for each analysis is also included (p.5 ll. 88-96, p.7 ll. 143-150).

Results: please adjust your P-values for multiple comparisons

Authors appreciate this important advice. We adjusted our P-values for multiple comparisons, applying the Bonferroni correction (p.7 ll.143-144); as a result, the association between FCV-19S-J and living place in Factor 1 and Factor 2 is no longer significant (p.11 l. 203).

Finally, following the advice of the academic editors, we have made the corrections to the text and added necessary information as follows.

1. We ensured that our manuscript meets PLOS ONE's style requirements, including those for file naming.

2. We provide additional details regarding participant consent (p.8, l. 153-155).

3. The lead author’s ORCID iD has been added and validated in Editorial Manager.

4. We added captions for our Supporting Information files at the end of our manuscript (p.27 l.500).

---

## [Editor Report · Decision Letter 1]

27 Jan 2021

Confirming Validity of The Fear of COVID-19 Scale in Japanese with a Nationwide Large-Scale Sample.

PONE-D-20-37753R1

Dear Dr. Tachikawa,

We’re pleased to inform you that your manuscript has been judged scientifically suitable for publication and will be formally accepted for publication once it meets all outstanding technical requirements.

Kind regards,

Chung-Ying Lin

Academic Editor

PLOS ONE
---

## [Editor Report · Acceptance letter]

1 Feb 2021

PONE-D-20-37753R1 

Confirming Validity of The Fear of COVID-19 Scale in Japanese with a Nationwide Large-Scale Sample. 

Dear Dr. Tachikawa:

I'm pleased to inform you that your manuscript has been deemed suitable for publication in PLOS ONE. Congratulations! Your manuscript is now with our production department. 

Kind regards, 

on behalf of

Dr. Chung-Ying Lin 

Academic Editor

PLOS ONE